# Dysfunctional High-Density Lipoproteins in Type 2 Diabetes Mellitus: Molecular Mechanisms and Therapeutic Implications

**DOI:** 10.3390/jcm10112233

**Published:** 2021-05-21

**Authors:** Isabella Bonilha, Francesca Zimetti, Ilaria Zanotti, Bianca Papotti, Andrei C. Sposito

**Affiliations:** 1Atherosclerosis and Vascular Biology Laboratory (AtheroLab), Cardiology Department, State University of Campinas (Unicamp), Campinas 13084-971, Brazil; bella_bonilha@hotmail.com; 2Department of Food and Drug, University of Parma, 43124 Parma, Italy; ilaria.zanotti@unipr.it (I.Z.); bianca.papotti@unipr.it (B.P.)

**Keywords:** high density lipoprotein, type 2 diabetes mellitus, HDL function, glycation, oxidation, antioxidant, anti-inflammatory, vasodilator, cholesterol efflux capacity

## Abstract

High density lipoproteins (HDLs) are commonly known for their anti-atherogenic properties that include functions such as the promotion of cholesterol efflux and reverse cholesterol transport, as well as antioxidant and anti-inflammatory activities. However, because of some chronic inflammatory diseases, such as type 2 diabetes mellitus (T2DM), significant changes occur in HDLs in terms of both structure and composition. These alterations lead to the loss of HDLs’ physiological functions, to transformation into dysfunctional lipoproteins, and to increased risk of cardiovascular disease (CVD). In this review, we describe the main HDL structural/functional alterations observed in T2DM and the molecular mechanisms involved in these T2DM-derived modifications. Finally, the main available therapeutic interventions targeting HDL in diabetes are discussed.

## 1. Background

Diabetes mellitus (DM) is a chronic metabolic disorder characterized by defective insulin secretion and reduced tissue sensitivity to insulin, leading to hyperglycemia. Mortality and morbidity resulting from DM are consequences of long-term micro and macrovascular complications. The increasing prevalence of DM worldwide represents an important public health problem. The latest edition of the International Diabetes Atlas showed that 463 million adults live with DM and that more than 4 million people aged between 20 and 79 years died from causes related to DM in 2019. In addition, estimates show that there will be 578 million individuals affected by 2030 and 700 million in 2045 [1].

Type 2 diabetes mellitus (T2DM), characterized by an increase in blood glucose as a result of progressive increase in insulin resistance and pancreatic incompetence to meet the progressive demand for insulin production, corresponds to about 90% of all cases of DM and its prevalence is increasing worldwide. T2DM and insulin resistance are also well known for significantly increasing cardiovascular (CV) risk. Notably, in individuals with T2DM, an increase of 2 to 4% in mortality from coronary artery disease is estimated [2]. The dysfunction of pancreatic beta cells, a critical component for the pathogenesis of T2DM, has been attributed to glucotoxicity and high levels of free fatty acids with a high inflammatory response. An additional possible pathogenic mechanism is that hyperglycemia accelerates atherogenesis by increasing the oxidation of lipoproteins.

According to Kashyap, levels considered normal or even high for high density lipoproteins (HDLs) in the diabetic individual are not equivalent to its appropriate functionality [3]. So, as T2DM progresses, changes occur in the composition of HDL, and, as a consequence, these particles lose their biological activities becoming a dysfunctional lipoprotein. In this review, we focused on the discussion of HDL dysfunction in T2DM.

## 2. High Density Lipoproteins

HDLs, first described in 1921 [4], are endogenous particles resulting from a complex interaction of proteins and lipids organized by a single layer of amphipathic molecules on their surface and, therefore, are capable of transporting hydrophilic molecules on its surface and hydrophobic molecules inside. HDL biogenesis starts from nascent particles that consist mainly of apolipoprotein A-I and apolipoprotein A-II (ApoA-I and ApoA-II, respectively) and phospholipids, which are secreted by the intestine and the liver. As the nascent particles circulate through the intravascular environment, they acquire phospholipids, numerous proteins (up to 96), cholesterol, and other lipids and microRNAs, becoming spherical particles [5,6]. Among the newly acquired apolipoproteins, there are members of the apolipoprotein C (ApoC) family. In particular, ApoC-I is a potent activator of lecithin-cholesterol acyltransferase (LCAT), the enzyme responsible for the esterification of cholesterol and the remodeling of small HDLs towards mature particles [7]. Other enzymes primarily carried in the bloodstream by HDLs are paraoxonases (PONs) and the platelet-activating factor acetyl hydrolase (PAF-AH). The former are calcium-dependent lactonases expressed as three isoforms. The major physiologic function of PON1 is the hydrolysis of homocysteine thiolactone [6]. The latter is a Ca^2+^-independent phospholipase A_2_, which catalyzes the conversion of platelet-activating factor (PAF) by hydrolysis of the acetyl group at the sn-2 position of the glycerol structure [8].

Traditionally, ultracentrifugation was the first method applied to the isolation of lipoproteins [9,10]. This technique allowed the identification of HDL heterogeneity in human plasma. Most HDL particles have a density between 1.063 and 1.21 g/mL. More recently, through sequential ultracentrifugation, two main subfractions were isolated, HDL_2_ (d = 1.063–1.125 g/mL) and HDL_3_ (d = 1.125–1.21 g/mL) [11]. Based on the size, three classes of particles can be distinguished: large HDLs (8.8–13.0 nm in diameter), medium HDLs (8.2–8.8 nm), and small HDLs (7.3–8.2 nm). Furthermore, HDL_2_ and HDL_3_ subclasses have been identified: HDL_3c_, 7.2–7.8 nm in diameter; HDL_3b_, 7.8–8.2 nm; HDL_3a_, 8.2–8.8 nm; HDL_2a_, 8.8–9.7 nm; and HDL_2b_, 9.7–12.0 nm [6].

Importantly, HDLs are subject to constant intravascular remodeling that generates multiple HDL phenotypes. The predominant characteristics of HDLs for a given individual will depend on the activity of remodeling enzymes and the exposure of HDLs to that individual’s intravascular and extravascular milieu [12]. This complex interaction can promote differences in protein composition, in particle diameter, in the degree of oxidation or glycation of lipids and proteins, and in the proportion between phospholipids. As we discuss later, all these modifications implicate significant alterations of the well-recognized beneficial functions of these lipoproteins.

## 3. Alterations of HDL Plasma Level and Composition in T2DM

It is well established that T2DM is associated with quali-quantitative changes in circulating lipoproteins. The lipoprotein profile of dyslipidemic, diabetic subjects includes an increase in triglycerides (TGs) and apolipoproteinB (ApoB)-containing lipoproteins (especially small and dense low-density lipoproteins (LDLs) and very low-density lipoproteins (VLDLs)) and a decrease in HDL levels. In addition, HDL structure and composition are significantly perturbed, with abnormal enrichment in TGs and depletion of cholesterol and ApoA-I [13]. Thus, the phenotype of HDL in T2DM is the result of both reduced levels of circulating particles, and alterations, either as an increase or decrease, in crucial structural components. In the following paragraphs, we detail the compositional and functional characteristics of these abnormal HDLs (Figure 1).

Schematic representation of the main changes in HDL composition due to T2DM. HDL size of diabetic patients is altered, with loss of large and very large HDL_2_ and with a shift toward small HDL_3_. Accordingly, the lipid core is modified, with an enrichment in triglycerides (pink) instead of CE [14], resulting in less stability and higher renal elimination [15]. As a consequence of the reduction in surface lipids (phosphatidylcholine, ether-linked phosphatidylcholine, sphingomyelin, ceramides and free cholesterol), HDLs in diabetes are characterized by an alteration of their architecture and fluidity. Additionally, the content of proteins in HDLs is significantly altered in T2DM, with an increased SAA, fibrinogen, ApoC-II, and ApoC-III levels, and a reduction in ApoA-I, ApoA-II, ApoE, ApoM, and PON-1. Moreover, because of hyperglycemia and oxidative stress, diabetic patients present a large array of glycosylated proteins [16]. All these compositional changes have a significant impact on the antiatherogenic functions of HDL, resulting in impaired antioxidant, anti-inflammatory, and vasodilator activity, together with a reduction in HDL cholesterol efflux capacity in patients with T2DM [17,18,19,20].

### 3.1. Modifications of HDL-C Plasma Levels

Dyslipidemia is a peculiar feature of individuals affected by T2DM and it is observed in 60–70% of patients [21].

Despite the observation that the appearance of low levels of HDL-cholesterol (HDL-C) levels may precede the onset of DM, the direction of the causal relationship between diabetes and reduced HDL is still a matter of debate. The hypothesis of low HDL-C levels as a consequence of diabetes is supported by the evidence that, in a setting of insulin resistance, elevated plasma TGs may drive a cholesteryl ester transfer protein (CETP)-mediated mechanism leading to the formation of cholesteryl ester (CE)-depleted, small HDL that are rapidly catabolized by the kidneys [15]. On the other hand, pre-existing low HDL-C levels may facilitate the onset of diabetes and its complications through the loss of protective functions on pancreatic beta cells [22] or on endothelial cells [23]. Epidemiological studies have consistently shown that low plasma levels of HDL-C are inversely related to the risk of T2DM development [24,25] and that they are independent predictors of diabetes complications such as the amputation of lower extremities, wound-related mortality [26], and diabetic nephropathy [27].

This controversy is not resolved by genetic studies, which provide conflicting results on the relationship between HDL and T2DM. While some reports suggest that a genetic predisposition to low HDL predicts an elevated risk of T2DM [28], others, based on a Mendelian randomization approach, did not support such an association [24].

### 3.2. Modifications of HDL Size

Seminal studies employing density ultracentrifugation techniques revealed significant perturbations of HDL size in diabetic individuals, with the loss of large and very large HDL_2_ and the gain of small HDL_3,_ rich in TGs and poor in cholesterol [29]. Consistently, two-dimensional gel separation studies have shown that diabetic subjects exhibit lower levels of large α-1, α-2, and pre-α-1 particles and higher levels of lipid-poor α-3 HDLs [30]. Using the nuclear magnetic resonance (NMR) spectroscopy method, it has been demonstrated that subjects with diabetes present reduced levels of medium (9.0–11.5 nm) and large (11.5–18.9 nm) HDL particles but enriched small (7.8–9.0 nm) HDL compared to controls [31]. Interestingly, it seems that particle size shifts in HDLs may precede the diagnosis of T2DM, as suggested in a study that found that levels of small HDL particles were positively associated with future T2DM risk, whereas large HDL particles showed an inverse association [32]. This remodeling is mainly driven by the above-mentioned raise in CETP activity, leading to increased production of TG-enriched HDLs. These particles represent the optimal substrate for endothelial and hepatic lipases that promote the production of small, dense HDLs [33]. In addition, the shift in HDL size can also be attributed to impaired LCAT activity due to high levels of glycated HDL (see below), which represent poor substrates for this enzyme [34,35].

In the context of HDL size changes, it is worth remarking that a “gold standard” separation method for HDL subclasses is still lacking [36] and that further studies are needed to univocally conclude on which subclasses of these particles are most related to their atheroprotective functions, even in the T2DM setting.

### 3.3. Modifications of the Lipid Content

As previously cited, the increase in TGs and decrease in CEs in HDLs is a hallmark of T2DM, and it has been established for years. This lower CE/TG ratio, by conferring less stability than normal particles, is likely to be responsible for the higher susceptibility to renal elimination [37].

In recent years, the application of mass spectrometry and NMR-based techniques has allowed the full characterization of the HDL lipidome in health and disease. In particular, two studies [38,39] showed consistent results on plasma lipid distribution in dyslipidemic, diabetic individuals and demonstrated that most species of lipids were present in significantly lower concentrations, whereas only a minority was increased compared to HDLs from control subjects [38]. In detail, surface lipids such as phosphatidylcholine, ether-linked phosphatidylcholine, sphingomyelin, ceramides, and free cholesterol (FC) were reduced between 10 and 50% in individuals with T2DM and dyslipidemia [39]. Since surface lipids are components that determine the architecture of HDL particles, such as the degree of surface fluidity, it is expected that these alterations may impair the properties related to their function as an acceptor of cholesterol and anti-inflammatory or anti-oxidant activity (see Section 4.1 and Section 4.2) [13,39]. In addition, the lipid content of the HDL core showed a reduction in CEs (−8%) and an increase in the absolute amounts of TGs and diacylglycerol (DG) (+77%) in diabetic compared to control particles [39].

Interestingly, these works reported discrepant results on lysophosphatidylcholine (LPC) content. Whereas Ståhlman and colleagues found increased LPC in HDL from dyslipidemic, diabetic subjects [39], in Cardner’s work, it was reduced [38]. This difference is of potential interest for the implications on the inflammatory properties of HDL. In fact, as it is well known, LPC is constituted by fatty acids that can be either anti-inflammatory (polyunsaturated fatty acids) or pro-inflammatory (arachidonic acid). In Ståhlman’s work, the observation that LPC is mainly associated with arachidonic acid suggests that small, dense HDLs may represent a biomarker of the inflammatory milieu in diabetes. Additionally, Ståhlman and colleagues have noted that atherogenic dyslipidemia, and not insulin resistance or hyperglycemia, is the main driver of HDL lipidome perturbations. In fact, the above described alterations were not observed in HDL isolated from normolipidemic, diabetic people [39].

The reduction in sphingosine 1-phosphate (S1P) content of HDLs has been observed in diabetic patients [40], with negative implications on HDL vasodilator activities (see Section 4.3).

### 3.4. Modifications of the Protein Component

Most studies agree with the observation that the levels of ApoA-I, the most prominent protein of HDL, are significantly reduced in T2DM. Multiple mechanisms are likely to account for this modification: (i) the affinity of ApoA-I for the small HDL particles typical of T2DM is reduced, leading to the dissociation of ApoA-I and the consequent accelerated clearance by the kidneys; (ii) the synthesis of ApoA-I may be reduced through a mechanism of inhibition of transcription factors driven by high glucose levels; (iii) the ApoA-I expression is reduced as a consequence of the insulin resistance [41]; (iv) the binding of the pro-inflammatory protein serum amyloid A (SAA) to HDL is accompanied by the removal of ApoA-I [37].

In general, the content of proteins in HDL is significantly altered in T2DM, as comprehensively illustrated in a recent study assessing the proteome of HDL from diabetic and healthy subjects. This work revealed that 17 proteins were increased and 44 were decreased in the disease status. Among the former, it is worth mentioning SAA, fibrinogen, ApoCII, and ApoCIII. Among the latter, relevant examples were ApoA-IV, apolipoprotein E (ApoE), apolipoprotein M (ApoM), and paraoxonases (especially PON1) [38] (Figure 1). The reported increase in plasma levels of ApoC-II and ApoC-III is consistent with the elevated TGs detected in diabetic subjects [42]. Since glucose induces ApoC-III transcription, a mechanism that links hyperglycemia and hypertriglyceridemia in patients with T2DM has been suggested [43]. Indeed, a recent multi-ethnic study demonstrated an adverse association between increased plasma ApoC-III levels and the risk of diabetes, while HDLs lacking ApoC-III were associated with lower incidence. These observations may suggest that ApoC-III negatively impacts HDL functions, probably by disrupting glucose homeostasis and enhancing circulating triacylglycerol levels [44]. Consistent results were observed when HDL from adolescents with T2DM were analyzed. In this young population the typical feature of atherogenic dyslipidemia is apparent, with a significant increase in plasma TGs and a decrease in HDLs, which are also shifted to smaller particles. Interestingly, a significant reduction in all proteins, including ApoA-I, ApoA-II, ApoE, ApoM, and PON1 was found [45]. These findings suggest that such changes represent early markers of the disease. A summary of these HDL phenotypes and functional changes in T2DM is displayed in Table 1.

### 3.5. Modifications of HDL Due to Glycation and Oxidation

Diabetic patients are under the conditions of hyperglycemia and oxidative stress. It is well known that the long-term exposure of proteins and lipids to high levels of glucose leads to the non-enzymatic glycation of several macromolecules, of which the HDL is a primary example [46]. Simultaneously, oxidative stress may modify HDLs through multiple mechanisms, as demonstrated by the increase of 127% of oxidation as measured by the TBARS assay [47], or the nitration of ApoA-I [48]. Thus, collectively, these processes affecting HDL composition and function may be termed “glycoxilation” [16].

The first evidence of non-enzymatic glycation of HDL in vivo was published in 1985. Curtiss and Witztum verified glycation in plasma lipoproteins of diabetic individuals using an innovative immunochemical approach that employed monoclonal antibodies recognizing glycosylated residues on proteins. The results showed that diabetic patients presented a large array of glycosylated proteins, including ApoA-I, ApoA-II, ApoB, ApoC-I, ApoE and albumin, primarily in HDL. Of note, the relevance of a high extent of glycosylation in TG-rich lipoproteins of diabetic individuals was explained by the transfer of glycosylated apoproteins from HDLs [49]. Although this study has been carried out in only three hyperlipidemic, diabetic individuals with poor glucose control, the increased extent of HDL glycation has been later confirmed in larger cohort of T2DM subjects, and it has been related to plasma levels of glucose [50]. Recently, an elegant work of proteomics comparing plasmas from control and diabetic individuals revealed that the latter present significantly higher glycated ApoA-I, suggesting that this parameter could be considered as valuable diagnostic tool to assess the metabolic state of diabetic patients that experience a glyco-oxidation stress during the half-life of the protein [51].

Moreover, an in vitro study by Hedrick and collaborators showed that HDLs incubated with high concentrations of glucose had a fourfold increase in glycation, especially on the protein component, including ApoA-I, paraoxonases, and malondialdehyde, an important marker of oxidative stress [16].

## 4. Alterations of HDL Functions in T2DM

The above described compositional changes that HDLs undergo in T2DM have a significant impact on the antiatherogenic functions of these lipoproteins. In this section, we examine the most relevant reports documenting alterations of the anti-inflammatory, antioxidant, vasodilator properties of HDL occurring in T2DM, as well as modifications of the capacity to promote cholesterol efflux, the first, limiting step of the reverse cholesterol transport (RCT) process. We also discuss the potential mechanisms underlying these modifications.

### 4.1. Impaired Antioxidant Activity

The increased release of reactive oxygen species (ROS) associated with diabetic dyslipidemia and hyperglycemia is one of the important events leading to endothelial dysfunction, atherogenesis, and CV disease (CVD) in diabetic patients. Moreover, the increased formation of modified lipid species deriving from the oxidation of arachidonic and linoleic acid because of the increased inflammatory status and oxidative stress has been responsible for functional changes in HDLs observed in T2DM [19].

Considerably, HDLs from diabetic patients had no significant inhibitory effect on the production of endothelial cell superoxide or on nicotinamide adenine dinucleotide phosphate (NADPH) oxidase activity, indicating a loss of these HDL antioxidant effects on the endothelium. In the HDLs of diabetic patients, a substantial increase in lipid peroxidation was observed through the measurement of malondialdehyde and by electrophoresis mobility assay. In addition, serum myeloperoxidase levels are independently associated with endothelial dysfunction [20].

As anticipated above, previous studies have shown that activity of the HDL-associated antioxidant enzyme PON1 is reduced in individuals with T2DM [63]. The low serum PON1 activity, beyond reducing the antioxidant action of HDL, is associated with microvascular alterations and increases lipid peroxidation and contributes to diabetic complications and mortality [62]. In particular, the relationship between PON1 and diabetes is peculiar and reciprocal, with diabetes reducing PON1 levels on the one hand and the PON1 genotype being associated with the risk of diabetes development on the other [64].

A possible explanation for the reduction in PON1 activity in diabetes is the displacement of PON1 by SAA, due to the inflammatory status and oxidative stress. In fact, in HDLs, PON1 physiologically associates with ApoA-I, this interaction being important for the optimal PON1 activity/stability [65]. Consistent with this observation, a cross-sectional study comparing T2DM patients and healthy controls revealed a loss of the relationships of PON1 with HDLs and ApoA-I levels, potentially leading to HDL dysfunction [55]. Another possible explanation for reduced PON1 activity is related to an increased glycation [66]. In this regard, Hedrick showed that glycated HDLs present a 65% reduction in the PON1 enzymatic activity [16]. Glycation of HDLs has also been linked to the pro-atherogenic activity on vascular smooth muscle cells through the formation of ROS. In particular, in vitro glycated HDLs as well as HDLs isolated from T2DM patients have been demonstrated to induce the proliferation and migration of vascular smooth cells, and this effect was abrogated by ROS suppression. These data clearly suggest how these diabetes-induced modified lipoproteins might contribute to the progression of atherosclerosis [67].

Among other possible mechanisms, the activity of the PAF-AH was also implicated, since it was found to be reduced in the HDL_3C_ fraction of diabetic patients, contributing to impaired antioxidant action [68]. Notably, the small and dense subfractions of HDLs of diabetic patients, mainly the HDL_3B_ and HDL_3C_ fractions, have shown a reduction by up to 52% in protecting LDLs against oxidative stress [68]. Conversely, in a work assessing the activity of HDL subfractions against LDL oxidation, it was noted that large HDL_2_ from diabetic subjects exhibited decreased protection against LDL oxidation compared to healthy controls, in association with decreased FC and increased TGs [13]. Additionally, it is worthwhile to mention that the reduced antioxidant capacity of HDLs in diabetes may also end up in increased LDL pro-oxidant properties, as demonstrated by Morgantini et al. in a cross-sectional study comparing 93 diabetic patients and 31 healthy subjects [19].

### 4.2. Impaired Anti-Inflammatory Activity

The changes in HDL protein composition and size induced by T2DM may impact the anti-inflammatory properties of these particles. The prominent role of this protective activity in diabetes is well explained in a recent work where the anti-inflammatory and the antiangiogenic properties of HDL have been reviewed for their potential beneficial role in diabetes-related wound healing [69].

The ability of HDLs to mitigate the inflammatory response is significantly reduced in T2DM, even in patients with sufficient metabolic control. Specifically, Ebtehaj found a 3.18-fold increase in *vascular cell adhesion protein 1 (VCAM-1)* mRNA expression in endothelial cells exposed to HDLs from diabetic compared to non-diabetic participants. This effect was associated with an increase in plasma levels of high sensitivity C-reactive protein (CRP) and tumor necrosis factor-α (TNF-α) [63].

Among the mechanisms explaining the link between HDLs, inflammation, and diabetes, Ebtheai and colleagues have pointed to the glycation process. The authors found a significant correlation between the reduced HDL anti-inflammatory activity and hyperglycemia. According to this hypothesis, it has been reported that glycation of HDLs in vitro reduced their anti-inflammatory properties on adipocytes in which inflammation was triggered by palmitate treatment [18]. Consistently, the in vitro glycation of HDL caused the loss of the ability to suppress TNF-α and interleukin-1β (IL-1β) production by lipopolysaccharide (LPS)-stimulated macrophages [70]. Thus, the progression of inflammation in diabetes plus the change in HDLs due to hyperglycemia together reduce the anti-inflammatory and antioxidant properties of HDLs and its associated proteins such as ApoA-I [53]. Another proposed mechanism explaining the loss of anti-inflammatory activity of HDLs is their enrichment in SAA, as mentioned in the sections above [18]. Accordingly, increased SAA levels in diabetic nephropathy were directly associated with HDLs’ impaired capacity to reduce the secretion of TNF-α from LPS-treated monocytes [71]. Mechanistically, Vaisar and colleagues showed that the HDLs of diabetic patients were not able to suppress the TNF-α dependent activation of the nuclear factor κB (NF-κB) in human endothelial cells [25]. Moreover, glycated HDLs are not able to inactivate oxidized LDLs, resulting in increased monocyte adherence to endothelial cells [16].

The size of HDLs is likely to be an additional factor contributing to the pro-inflammatory properties of these particles in diabetes. In fact, it has been demonstrated that the predominant small HDLs are primary carriers of ceramides, which are recognized as potent activators of the transcription factor NF-κB [39].

### 4.3. Impaired Vasodilator Activity

Among the several antiatherogenic properties, HDLs display endothelial protective functions through the ability to stimulate nitric oxide (NO) production. This capacity may be impaired in many clinical conditions including diabetes. As an example, it was shown that the HDLs of diabetic patients displayed a reduction by up to 40% in the ability to stimulate the phosphorylation and activation of endothelial NO synthase (eNOS)^Ser1179^, the enzyme responsible for NO production [25]. Indeed, Sorrentino also showed that HDLs from diabetic patients lose the capacity to stimulate NO production from endothelial cells, as detected by spectroscopy analysis. Moreover, the diabetic HDLs are not able to stimulate the endothelium-dependent vasodilation [20].

The association between the modifications of HDL activity induced by diabetes and vascular dysfunction is further highlighted by the results of a recent work on 125 patients with T2DM [72]. The authors found that the levels of plasma-nitrated HDLs, resulting from the myeloperoxidase (MPO)-induced modification of ApoA-I, were a predictor of vascular dysfunction [73]. The authors also observed that this link was independent of total NO availability, suggesting an NO-independent pathway. As seen for other HDL properties, glycation may affect the vasodilation ability of HDLs. Glycation of HDL may be responsible for the reduced S1P binding to HDL [74], with a consequent loss of the S1P receptor-mediated activation of eNOS and the release of NO [75].

Concerning the HDL impact on other lipoproteins, modified HDLs in T2DM lose the ability to protect endothelium from oxidized LDL-induced inhibition of vasorelaxation. Again, the inhibitory effect of oxidized LDLs was correlated with the TG content of HDLs [58].

### 4.4. Impaired Cholesterol Efflux Capacity

The cholesterol efflux capacity (CEC) is undoubtedly the functional property of HDLs that has been mostly studied in relation to CV risk because of its strong association with atherosclerosis prevalence and CVD incidence, as has emerged from several cross-sectional and longitudinal studies [76,77,78]. Importantly, this association appears to be independent of HDL-C plasma levels.

In the case of T2DM, the relationship with HDL CEC appears quite complex and the underlying mechanisms causing HDL dysfunction in T2DM are far from being fully clarified. In particular, in most studies, HDL CEC from T2DM patients was found to be reduced compared to healthy subjects [17,79]. Notably, in a cohort of about 600 T2DM patients and non-diabetic controls matched for plasma HDL-C levels, a reduction in the capacity of HDL to promote cholesterol efflux through the transporter ATP-binding cassette A 1 (ABCA1) was observed in the former [80]. In support of this data, very recently, by isolating HDLs by size from diabetic subjects, He Y. and colleagues were able to demonstrate a specific impairment of the capacity of the small HDLs, the subspecies predominant in diabetes (see Section 3.2) to promote cholesterol efflux via the transporter ABCA1. The authors also explain this impairment with a loss of a specific HDL-associated protein, the serpin family A member 1 (SERPINA1), since the enrichment of small HDLs with this protein was able to restore CEC [81].

Interestingly, a 30% reduction in CEC was detected not only in plasma HDL from diabetic subjects but also in particles isolated from the interstitial fluid, the peripheral compartment where the RCT begins [82]. Consistent with the hypothesis of an impaired CEC in T2DM, Blanco-Rojo R. and colleagues demonstrated in a prospective setting that HDL CEC normalized to ApoA-I levels was inversely associated to the future development of T2DM in a cohort of more than 400 subjects free from the disease at the baseline [54]. Such an association persisted after adjustments for glucose and glucose metabolism-related parameters, strengthening the importance of HDL function not only in the context of CVD but also in other chronic diseases such as T2DM.

As for the other HDL functions examined above, the impaired CEC may be caused by the glycoxidation of ApoA-I, as previously reported [3,83]. Zheng et al. demonstrated direct evidence that ApoA-I is the preferred target for nitration of tyrosine residues and chlorination catalyzed by MPO, which is negatively correlated with the ABCA1-dependent cholesterol efflux in macrophages, generating pro-atherogenic HDLs [56]. A potential role of ApoA-I autoantibodies, whose levels are increased in T2DM patients with CVD [84], has also been suggested, since their levels are inversely correlated with HDL CEC. This association was proven to be independent of pre-β-HDL formation and of the levels of remodeling enzymes acting on HDLs, but it was lost when the plasma cholesterol esterification was introduced in the model [84]. This evidence suggests a complex relationship between the efflux and the esterification process in T2DM that probably involves the exchange of CEs between ApoB-containing lipoproteins and HDLs [85].

Despite the reported data, the literature is not univocal in associating T2DM to impaired HDL CEC, and in contrast with the above works, other studies found either no changes [86,87] or increased CEC in diabetics compared to healthy subjects [88]. The increased CEC was presumably a consequence of a shift in HDLs toward small particles, as discussed in Section 3.2. In another work, it was instead demonstrated that cholesterol efflux from macrophages was well correlated with medium-sized HDLs, rather than with small HDLs, in patients with well-controlled T2DM, even after adjustment for ApoA-I [89].

The explanation for these incongruences among the studies may be related to the complex interplay among several factors, including the severity of the disease, the degree of hyperglycemia, as well as the pharmacological treatments that these subjects undergo, which may potentially influence HDL CEC. In addition, the cellular system utilized to assess HDL CEC can affect the results. HDLs are a heterogeneous class of particles, with each sub-particle preferentially interacting with specific cholesterol efflux transporters [90]. Depending on the expression of such transporters on the cell membrane, one effect or the other may predominate, making it difficult to establish a net impact on the global cholesterol efflux capacity.

To summarize, several reports have demonstrated an impairment of HDL CEC, mostly due to modifications of specific HDL subfractions or glycation of its main apolipoprotein component. In addition, it needs to be taken into consideration that diabetes is also associated with a reduced macrophage expression of ABCA1 and ATP-binding cassette G1 (ABCG1), the major membrane transporters involved in RCT [91,92]. In light of these observations, we may thus imagine a scenario with a double impairment of the RCT process in this metabolic disease, where the impaired release of cholesterol from cells of the arterial wall is coupled with a reduced capacity of extracellular particles to discharge excess cholesterol. This double defect is likely to have a significant detrimental impact on atherogenesis in diabetic patients.

## 5. Therapeutic Interventions

Several epidemiological studies in humans have established an association between plasma HDL-C levels and the risk of developing coronary artery disease. Gordon et al., [93] estimated that a 1 mg/dL increase in HDL was associated with a significant 2–3% reduction in the risk of heart disease. Although the therapies used to increase plasma HDL levels failed in the reduction in CV events [94], in some cases, there was an improvement in glycemic control in diabetics [52,95]. Some lipid-lowering drugs, such as niacin and fibrates, have a mixed pharmacological profile making them capable of increasing plasma HDL levels. In addition, sodium–glucose cotransporter type 2 inhibitors, widely employed in T2DM subjects, may potentially exert pleiotropic effects on HDLs. In the following paragraphs, we will summarize data reporting the capacity of these drugs to target HDLs in patients with T2DM.

### 5.1. Niacin

Niacin or nicotinic acid was the first oral drug available to treat high cholesterol levels and the first description of its hypolipidemic effects was in 1955 [96]. Niacin is one of the most effective compounds in increasing plasma HDL-C, whose levels are raised by up to 35% [97,98]. It has been suggested that niacin’s ability to increase HDL-C levels is mediated in part by an increase in ApoA-I production [99], likely related to the induction of ABCA1 expression [100]. Other explanations include an inhibitory effect on CETP [99] or a reduced HDL catabolism [101]. As a consequence of CETP inhibition, niacin treatment also affects the HDL size, by increasing the number of medium and large particles [102]. However, the impact on HDL CEC, specifically on the pathways involving large HDLs (i.e., scavenger receptor class B type 1 (SR-BI), ABCG1, and aqueous diffusion), is overall modest or absent across all the studies [103,104]. This inconsistency extends to niacin’s relationship with CV risk reduction as reported by both the AIM-HIGH investigators and the HPS2-THRIVE collaborative group, who both demonstrated no additional CV clinical benefit in adding niacin to statin therapy [105,106], ruling out the role of niacin treatment in diabetic dyslipidemia.

Niacin is considered an effective drug for T2DM-associated dyslipidemia for its broad effects on lipids, including HDLs. Such effects in diabetic patients are explained by an increased ApoA-I concentration as a consequence of a reduced catabolic rate [107]. In particular, the raising effect on HDL-C levels in T2D patients emerged from a post hoc analysis of the lipid-modifying efficacy of extended release (ER) niacin/laropiprant. In this study, the efficacy on HDL-C was independent of the degree of the glycemic control of the diabetic patients at baseline [108]. The increase in plasma HDL-C levels in T2D patients was further demonstrated by a meta-analysis of seven studies; although it also highlighted a worsening of the fasting plasma-glucose associated with the treatment [109], suggesting that monitoring of glucose may be required in long-term treatments with niacin. Similarly, a more recent meta-analysis including 2110 subjects with T2DM across eight trials suggested that niacin significantly increased plasma HDL-C levels, except for the subgroup of subjects with follow-up ≥20.0 weeks [107]. Despite a positive effect on the lipid profile in this study, a potential harmful effect on plasma glucose and glycated hemoglobin (HbA1c) levels has been suggested.

Concerning the effects of niacin on HDL structure and function, the few results available in T2DM patients are controversial. For instance, a study examining the effect of niacin on 30 diabetic patients in a crossover-design setting showed that the treatment did not normalize the impaired HDL antioxidant function [110]. However, niacin tended to lower the levels of pre-β1-HDL and significantly decreased PAF-AH and PON-1 activities. Moreover, in a study on 33 T2D patients and 10 controls randomized to receive either ER niacin or placebo for 3 months, Sorrentino demonstrated that treatment with niacin, in addition to increasing plasma HDL-C levels, markedly improved endothelial protective functions in these patients, by promoting NO production, inducing the early endothelial progenitor cell-mediated repair, and by exerting antioxidant effects on endothelial cells [20]. These effects were associated with an improvement of patients’ flow-mediated dilation (FMD), suggesting niacin as potential vasoprotective agent. Thus, beyond the robust effects on HDL-C levels, further studies are needed to univocally conclude on impact of niacin on HDL functions in diabetes.

### 5.2. CETP Inhibitors

The first published information on a small molecule able to inhibit CETP occurred in 1994, after which a variety of inhibitors were synthesized and tested during the late 1990s [111]. The development of these molecules as novel cardioprotective drugs had a controversial course, with three compounds failing in phase III clinical trials with CV outcomes, despite the efficacy in raising HDL-C levels.

Torcetrapib (Pfizer, Gladstone, NJ, USA) is an orally administered selective inhibitor of CETP that efficiently increases serum levels of total HDL-C, the HDL_2_ fraction, and ApoA-I in a dose-dependent manner [112]. However, ILLUMINATE, a randomized trial enrolling 15,067 patients at high risk of CV, 88.7% of which had a history of T2DM, experienced an early termination because of the increased risk of deaths from CV (*n* = 49 vs. *n* = 35) and non-CV (*n* = 40 vs. *n* = 20) causes [113]. These fatal events occurred despite a 72.1% increase in HDL-C, a 24.9% decrease in LDL-C, and a 9% decrease in TG levels [113]. A post hoc analysis on 6661 patients from ILLUMINATE with T2DM confirmed that torcetrapib was effective in progressively raising HDL-C levels, reaching an increase of +66.8%, and improved diabetic control compared to subjects treated with only atorvastatin. Whether this effect was related to the increase in HDLs and their antidiabetic properties, as well as the impact of such an effect on CVD, has not been determined [95].

Dalcetrapib (Roche, Basel, Switzerland) differs from torcetrapib both in the structure and the mechanism of action unrelated to the formation of a CETP-HDL complex at therapeutic plasma concentrations [114]. Promising results on this compound derived from a post-hoc analysis including five placebo-controlled phase II clinical trials demonstrated a similar efficacy to reduce CETP activity, LDL-C, and TGs and to increase HDL-C and apoA-I in patients with and without T2DM [115]. Nevertheless, the OUTCOMES trial, enrolling 15,871 patients with acute coronary syndrome, 87% of which had elevated cardiac biomarkers at the time of the acute coronary event and 49% of which had T2DM, failed to associate the observed increase in HDL from 31 to 40% in the dalcetrapib group (compared to 4 to 11% in the placebo group) with a reduction in the risk of recurrent CV events [116].

Evacetrapib (Lilly, Indianapolis, IN, USA) is a potent CETP inhibitor with effects similar to torcetrapib. The results of the ACCELERATE trial [117] showed that patients who received evacetrapib had an average percentage increase in HDL-C levels of 133.2%, compared with an average percentage increase of 1.6% observed in patients in the placebo group. On the other hand, the average level of LDL-C decreased by 31.1% in the evacetrapib group and increased by 6.0% in the placebo group. However, no beneficial effect on CV outcomes was observed with the treatment with evacetrapib [117]. Recently, an analysis of a subset of about 8000 diabetic patients enrolled in the ACCELERATE trial demonstrated a lack of improvement by evacetrapib treatment in the first occurrence of the composite endpoint, represented by CV death, myocardial infarction, stroke, revascularization, and hospitalization for unstable angina. Notably, the absence of effects on the clinical endpoint was revealed, despite early, sustained amelioration of the serum lipid profile (increased HDL-C and reduced LDL-C) and improvement of the glycemic profile [118]. In the attempt to investigate the impact of evacetrapib on atherogenic lipid parameters, the ACCENTUATE trial was carried out, enrolling 366 subjects, 50% of whom had T2DM. The addition of evacetrapib to the atorvastatin treatment, beside increasing HDL-C and ApoA-I levels, improved both the ABCA1 and non-ABCA1-mediated CEC of HDL However, the impact of T2DM on the drug response was not specifically investigated [119].

Anacetrapib (Merck, Kenilworth, NJ, USA) is the most recently developed CETP inhibitor, with promising efficacy and safety profiles. The DEFINE trial showed that 24 weeks of treatment produced an increase in HDL-C levels from 41 to 101 mg/dL compared to a change from 40 to 46 mg/dL in the placebo group and a decrease in LDL-C levels from 81 to 45 mg/dL compared to the placebo group where the change was 82 to 77 mg/dL. Anacetrapib treatment had an acceptable side effect profile, did not result in the adverse CV effects seen with torcetrapib, and reduced atherosclerotic CVD [120], which are reasons for using the drug in future studies re-evaluating the hypothesis that CETP inhibition is cardioprotective. Notably, in a post-hoc substudy of DEFINE including about 600 patients, anacetrapib’s ability to modulate CEC was evaluated [121]. This study provided evidence of a promoting effect of the drug on this HDL function, which is not affected by the diabetes status, but it was blunted in T2DM subjects with allele “2” of haptoglobin, a protein associated with HDL whose polymorphism has been associated with impaired HDL functions [122] and increased CV risk in T2DM patients [123].

Lastly, the REVEAL study evaluated the safety of adding anacetrapib to atorvastatin. Of the enrolled individuals, 88% had a history of coronary heart disease and 37% had T2DM [124]. The average HDL level was 43 mg/dL higher in the anacetrapib group than in the placebo group (a relative difference of 104%). Thus, the addition of anacetrapib to intensive statin treatment in patients with atherosclerotic CV disease resulted in a lower incidence of major coronary events [125]. Consistent with data obtained with torcetrapib and evacetrapib, treatment with anacetrapib was associated with a lower incidence of new cases of diabetes.

### 5.3. Peroxisome Proliferator-Activated Receptor (PPAR) Agonists and Fibrates

Fibrates were first introduced in clinical practice in the late 1960s with clofibrate, which was subsequently dismissed because of excess mortality, and then followed by gemfibrozil, fenofibrate, fenofibric acid, bezafibrate, etofibrate, and ciprofibrate [126]. Fibrates are ligands of the peroxisome proliferator-activated receptor-α (PPAR-α) [127], which, once activated, heterodimerizes with the retinoid X receptor (RXR) and interacts with PPAR-response regulatory elements (PPRE), being responsible for the modulation of several genes involved in lipid metabolism, inflammation, and adipogenesis [128].

The most prominent effects of fibrates are a decrease in plasma TG-rich lipoproteins (−30–50%), mainly due to increased fatty acid uptake and β-oxidation, lipoprotein lipase (LPL) transcription, and repression of (ApoC-III) transcription. This, in turn, inhibits LPL activity, leading to the final lipolysis of TG-rich lipoproteins, with the subsequent reduction in chylomicrons and VLDLs [129]. Besides, fibrates increase ApoA-I and apoA-II levels, probably because of the direct effect on hepatic production [60], which results in raised HDL-C levels. Concerning the effect of fibrates on diabetes, several clinical studies evaluated the impact of PPAR agonists, mainly fibrates, on CV outcomes, including HDL-related parameters, in diabetic subjects. The ACCORD lipid study evaluated whether fenofibrate (160 mg daily) alone or combined with simvastatin would reduce the risk of CVD in 5518 patients with T2DM. Despite no changes in the rate of fatal CV events recorded, a modest increase in HDL-C levels (+2% vs. simvastatin alone) was recorded [130]. Interestingly, Linz and colleagues recorded a paradoxical reduction in HDL-C levels among some participants of the ACCORD lipid trial who received both fenofibrate and TZD, raising some concerns about this potential idiosyncratic reaction, which should be clinically monitored [131]. Similarly to the ACCORD lipid trial, in the DAIS study, 418 subjects with T2DM were randomly assigned to either fenofibrate (200 mg daily) therapy or placebo for a minimum of 2 years. Fenofibrate was shown to increase HDL-C (+18%) and ApoA-I levels, together with an increase in medium and small HDL particles, in the absence of any changes in insulin and adiponectin levels [132]. The FIELD study randomized 9795 participants with T2DM without statin therapy for fenofibrate (200 mg daily) or placebo. Four months of fenofibrate therapy resulted in an increase in plasma HDL cholesterol (+5%) compared to placebo-treated patients, together with an increase in ApoA-I and ApoA-II by 3.9% and 28%, respectively [133]. Moreover, a subset of the FIELD study evaluated the effect of fibrates on HDL structure and function: 33 patients with T2DM treated with fenofibrate were selected among the FIELD cohort (17 of them with low and 16 with high homocysteine levels) and were matched with 14 subjects allocated to placebo. High homocysteine levels and fenofibrate did not modulate either plasma pre-β-HDL levels or HDL properties such as HDL CEC, serum PON-1 mass, and phospholipid transfer protein (PLTP activity) [134]. Conversely, a placebo-controlled randomized cross-over study on a small group of 14 men with T2DM investigated the effect of simvastatin (40 mg daily), bezafibrate (400 mg daily), or their combination, showing that bezafibrate monotherapy increased HDL CEC, with a more robust effect when combined with simvastatin, whereas antioxidant and anti-inflammatory properties of HDL were not affected by these therapeutic interventions [135]. Masana and colleagues evaluated the impact of different lipid-lowering drugs on HDL particle distribution and composition in 30 patients with T2DM randomly distributed into one group, receiving simvastatin (20 mg daily) and fenofibrate (145 mg daily), and the other receiving simvastatin (20 mg daily), niacin (2 g daily), and laropiprant for 3 months. Fenofibrate was not associated with changes in HDL-C or ApoA-I levels but induced an increase in ApoA-II and pre β-HDL compared to baseline levels and a reduction in HDL-ApoC-III levels. Moreover, fenofibrate was associated with a reduction in PON-1 and PAF-AH activity [110]. As a consequence of these changes, a pooled meta-analysis of six randomized placebo-controlled trials considering 11,590 patients with T2DM clearly showed that long-term treatment with fibrates strongly reduced the risk of non-fatal myocardial infarction (−21%), despite no significant reduction in the risk of all-cause mortality, cardiac mortality, and other adverse cardiovascular outcomes [136].

Given the relevance of lipid abnormalities in diabetic patients, dual PPAR-α/γ agonists may potentially be beneficial in subjects with diabetes. Within this context, the activity of aleglitazar was investigated in a randomized, double-blind, placebo-controlled clinical trial in 332 T2DM patients. Four aleglitazar doses (50 μg, 150 μg, 300 μg, or 600 μg daily) were tested, showing an increase in HDL-C levels in a dose-dependent fashion (up to +28%) [137]. The favorable raising effect on HDL-C levels was confirmed by the results of a recent meta-analysis of seven studies, although it also emerged how aleglitazar treatment was associated with a poor safety profile, overall overwhelming the benefits on the lipid and glucose profile [137].

Finally, a new selective PPAR-α agonist, pemafibrate, was recently synthesized, showing a stronger ability in HDL-C elevation and lower liver and kidney toxicity compared to fenofibrate [138,139]. A large-scale phase III trial upon pemafibrate, PROMINENT (ClinicalTrials.gov Identifier: NCT03071692, accessed on 21 March 2021), is currently evaluating the impact of pemafibrate in dyslipidemic patients with T2DM [140].

The limited data on the effect of fibrates on HDL function in T2DM hamper the drawing of definitive conclusions on this aspect.

### 5.4. Metformin

Since numerous epidemiological studies have shown an association between the degree of glycemic control and CVD in T2DM, it is important to determine the role of therapies affecting glycemic levels on CVD and CV risk factors. Metformin is currently the first-line pharmacological approach for T2DM patients. The drug reduces hepatic glucose output and peripheral insulin resistance and may be considered a liver insulin sensitizer. It can also be used in combination therapy with sulfonylurea or sodium–glucose cotransporter-2 inhibitors (SGLT2is) when its maximally tolerated dose does not allow it to achieve the glycemic target within 3 months of follow-up [141,142].

Metformin may also modulate numerous epigenetic processes, such as histone acetylation as well as histone and DNA methylation, with a mechanism dependent on the phosphorylation of AMP activated protein kinase (AMPK) [143]. Such influence on the epigenetic machinery may contribute to its primary hypoglycemic action but may also result in off-target effects, including the modulation of lipid metabolism and the atheroprotection.

In particular, metformin treatment has been shown to reduce CVD with an effect independent of the glycemic control but rather on weight loss induction, blood pressure lowering, and plasma lipid profile improvement [144,145]. In particular, metformin showed the ability to reduce TC [146], whereas the effect on HDL-C is still unconclusive, as suggested by a previous review of nine analyzed studies [146]. However, in a more recent study on 155 individuals with newly diagnosed T2DM, metformin treatment for 12 months led to a significant decrease in serum LDL-C and TGs, whereas HDL-C levels were increased, although in a dose-independent manner [147]. To the best of our knowledge, no data are available on the effect of metformin on HDL function in T2DM.

### 5.5. Glucagon-Like Peptide-1 (GLP-1) Receptor Agonists

Glucagon-like peptide-1 (GLP-1) is a member of an endogenous class of hormones synthesized by intestinal epithelial cells in response to food intake, inducing glucose-dependent secretion of insulin and suppression of glucagon secretion [148]. Since endogenous GLP-1 is degraded within ~2–3 min in circulation, various GLP-1 receptor agonists have been developed to provide prolonged in vivo action. In this regard, diraglutide has been approved in the USA and European Union to improve glycemic control in patients with T2DM [149].

As described for metformin, GLP-1 receptor agonists have been shown to play a role in reverting epigenetic modifications induced by T2DM, possibly preventing atherosclerosis and vascular diabetic complications [150,151].

A number of trials have shown significant reductions in major CV events after treatment with GLP-1 receptor agonists [152,153,154,155]. These results were further supported by a recent meta-analysis, in which GLP-1 receptor agonist treatment reduced major adverse CV events (MACE) by 12% [156].

Concerning the effects on CV risk factors, including lipids, a recent meta-analysis evaluated the effect of liraglutide on cardiometabolic factors in individuals with or without T2DM. Whereas a marked effect was seen on BMI among T2DM patients, no changes were observed on plasma HDL-C levels [157]. This result is in line with a previous meta-analysis, in which no effect by liraglutide on HDL-C was observed [158], and with the results of others GLP-1 agonists [159,160]. Similarly, a treatment with a novel combination of glucose-dependent insulinotropic polypeptide and GLP-1 receptor agonist did not result in amelioration of the HDL-C plasma profile [161]. As for metformin, data on the potential ameliorating effect of these drugs on HDL function are not yet available.

### 5.6. Dipeptidyl Peptidase-4 Inhibitors

The glucose-lowering activity of dipeptidyl peptidase-4 (DPP-4) inhibitors is related to the inhibition of the enzyme degrading GLP-1 and other incretin hormones, leading to increased insulin and decreased glucagon secretion [162]. Beyond their primary effect on glycemic control, DPP-4 may also influence the CV system through GLP-1-dependent and independent effects, including favorable actions on CV risk factors such as lipids, body weight, blood pressure, inflammation, and oxidative stress, as demonstrated in pre-clinical studies [163]. However, these compounds failed to show similar benefits in human trials [164]. Concerning the effects on HDL-C, data are controversial and further investigations are certainly needed. For instance, the N-ISM study compared the long-term efficacy of ipragliflozin or sitagliptin given in combination with metformin for 52 weeks. It was reported that only the pragliflozin group showed a significant raise in the plasma levels of HDL-C [165]. Conversely, other studies, supported by results of a meta-analysis [166] failed to highlight any effect on these lipoproteins by treatment with DDP-4 inhibitors [167,168]. Additionally, for these molecules, the impact on HDL function has not been investigated yet.

### 5.7. Sodium-Glucose Cotransporter Type 2 Inhibitors (SGLT2is)

SGLT2is prevent resorption of glucose from the proximal renal tubules without inducing insulin secretion. SGLT2 inhibitors can be used as a monotherapy or in combination with metformin [14]. They have positive effects in addition to glucose control, such as weight loss, lowering blood pressure, and reduction in serum uric acid, possibly resulting in CV protection [169]. In this regard, the EMPA-REG OUTCOME study showed that treatment with empagliflozin determined a reduction of 14% in the relative risk for the primary CV composite endpoint of death from CV causes, non-fatal myocardial infarction (MI), or non-fatal stroke compared to placebo in patients with T2D and established CVD. Moreover, the treatment was associated with a 35% reduction in the risk of hospitalization for heart failure [170]. Following the EMPA-REG OUTCOME, the CANVAS trial demonstrated a lower rate of MACE in T2DM patients treated with canagliflozin compared to placebo [171]. Conversely, results of the DECLARE–TIMI did not highlight significant differences in the MACE primary outcome but rather a significant decrease in the risk of the second primary outcome, CV death or hospitalization for heart failure [172].

Additionally, in the case of SGLT2is, the beneficial CV effect goes beyond what is expected by improving glycemic control, and an epigenetic modulation has been hypothesized, occurring by favorable changes in specific circulating microRNAs [173].

To clarify the inconsistent effect of these molecules on the lipid profile across the studies on diabetic patients, a recent and systematic meta-analysis of 48 randomized controlled trials was conducted. It was found that SGLT2is led to an overall significant increase in HDL-C while TG levels were reduced, making SGLT2is highly useful for the clinical management of dyslipidemia in diabetic patients [174]. The proposed mechanisms of action relate to the improvement in insulin sensitivity and insulin secretion, leading to a reduced hepatic synthesis and increased catabolism of TG-rich lipoproteins [175]

Beyond the lipid profile, a randomized phase IV clinical trial evaluated the effects of dapagliflozin on both the HDL-C plasma levels and the size and function of these lipoproteins in patients with T2DM. At the end of the 12-week treatment with dapagliflozin 10 mg/day, HDL CEC was reduced by 6.7 ± 2.4% in the dapagliflozin group and by −0.3 ± 1.8% in the placebo group. However, this difference was lost when adjusting the analysis for age and BMI. In addition, no changes occurred in HDL-C levels, HDL particle size, and PON1 activity [176]. Conversely, a specific increase in the large HDL_2_ subfraction was observed in a previous study on 40 diabetic patients treated with dapagliflozin [177].

In order to deeply investigate the impact of SGLT2is on HDL function, the EXCEED-BHS3 prospective randomized clinical trial is currently ongoing. The study will compare the effects of empagliflozin 25 mg/day alone or combined with evolocumab in T2DM patients. The secondary endpoint of the study, aimed to evaluate changes in endothelial function induced by treatment, includes the assessment of the effect on HDL function, in terms of CEC and antioxidant/anti-inflammatory activities [178].

## 6. Conclusions

HDL is a nanoparticle with anti-atherogenic, antioxidant, and anti-inflammatory properties, and numerous studies confirm that the HDLs of individuals with T2DM have impaired biological activities, mainly because of oxidation and glycation. Understanding all the mechanisms behind lipoprotein dysfunction can favor the development of future therapies targeting such dysfunction and decreasing the risk of CVD [179,180] in patients with T2DM. As detailed above, therapies designed to increase plasma levels of HDL-C, either by stimulating synthesis or by interfering with its intravascular remodeling, have failed to demonstrate clinical benefit, making it clear that HDL functionality has a greater importance for atherosclerosis prevention than does its plasma concentration.

Possibly, the cellular apparatus that participates in HDL functions, such as the expression of transmembrane transport proteins or proteins involved in lipid synthesis, is subjected to disorders that prevent or mitigate a large part of the protective effects of this particle. If so, the entire HDL system needs to be considered together as a therapeutic target for HDLs’ protective action to occur. Besides, new approaches to modify the molecular composition of HDL and its ability to intervene under specific clinical conditions remains unexplored—for example, the increase in S1P in patients with acute coronary syndrome. Various HDL and apo A-I mimetics have been created and tested in preclinical studies or phase I and II trials but have not yet been evaluated in clinical trials with hard endpoints. Although exactly 100 years have passed since the discovery of HDL, the way to using this tool to mitigate CV risk is still far from being understood, particularly in individuals with T2DM. In fact, negative results in clinical trials and studies with Mendelian randomizations intellectually challenge the entire scientific community and discourage new research and researchers. In spite of this, the high residual risk with current preventive therapies and the broad spectrum of protective effects that have been related to HDL makes moving forward in this line of investigation an indisputable goal.

From a practical point of view, no intervention to increase HDL-C plasma concentration or to change its function, to date, has proven effective in randomized clinical trials. Thus, HDL dysfunction and its reduced levels should be considered as causal elements in the severity of atherosclerotic disease in individuals with T2DM. HDL functionality tests are not yet systematized so that they can be used in clinical practice and are restricted to the research scenario. Nevertheless, new approaches to systematize these methodologies, such as the use of apo AI or HDL mimetics and the possibility of combined therapies that target both the HDL particle and the cellular apparatus responsible for their interaction, remain to be explored. For now, the extensive volume of knowledge accumulated over years about the interaction between HDL and T2DM paves the way for a pathophysiology of severe atherosclerotic disease in these patients. In the near future, it is possible that this set may bring new perspectives for the characterization of risk and intervention.

## Figures and Tables

**Figure 1 jcm-10-02233-f001:**
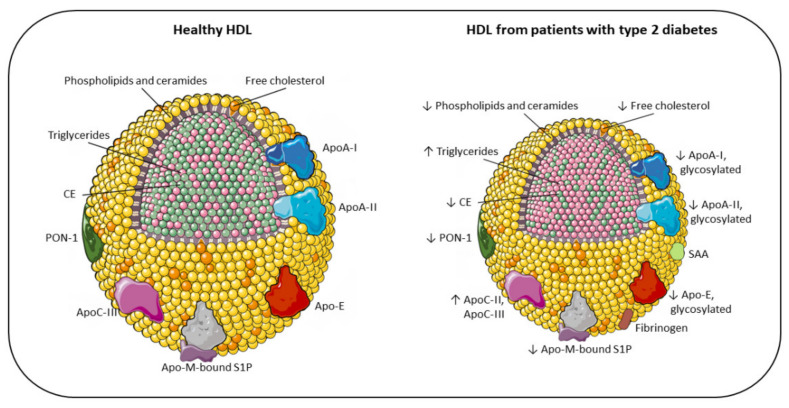
Main compositional differences between the HDL of healthy individuals and the HDL of patients with type 2 diabetes. Apo: apolipoprotein; CE: cholesteryl ester; PON-1: paraoxonase-1; S1P: sphingosine 1-phosphate; SAA: serum amyloid A; TG: triglyceride; 🡫: decreased; 🡩: increased.

**Table 1 jcm-10-02233-t001:** Summary of HDL’s structural and functional changes in patients with type 2 diabetes mellitus.

Changes in HDL Particle	Consequence in T2DM
**Alterations in Plasma Level and HDL Phenotype**	
🡫 HDL plasma level	Elevated plasma TGs [21,52]
Change in HDL size	Increase in small, dense TG-rich HDLs [32]
Lipid content	Increase in TGs and decrease in CEs, conferring less stability [14,38]
	Decrease in phospholipids and ceramides [38,39]
**Changes in HDL Components**	
🡫 ApoA-I	Changes in the size and composition of HDL [41]
	Glycoxidation of ApoA-I content, causing reduced LCAT activity [53]
	Impairs cholesterol efflux capacity and antioxidant activity [54,55,56]
🡫 ApoA-II	Change in the ability to bind to lipids [57]
🡫 ApoM-S1P	Reduction in the ability to bind to lipids due to an increase in hydrophilicity [40]
	Loss of NO release [58]
🡩 ApoC-II	TG increase in VLDLs and LDL due to impaired reverse cholesterol transport [59]
🡩 ApoC-III	Inhibition of LPL expression [60]
	Increased LDL susceptibility to hydrolysis [61]
	Inhibition of ApoE-dependent binding to the receptor [59]
🡫 PON-1	Loss of the ability to protect LDLs from oxidative damage [62]
	Reduction in cholesterol efflux [55]
	Proatherogenic HDL [55]
🡫 PAF-AH	Impaired antioxidant action [55]

Apo: apolipoprotein; CE: cholesteryl ester; LCAT: lecithin–cholesterol acyltransferase; LPL: lipoprotein lipase; NO: nitric oxide; PAF-AH: platelet-activating factor-acetylhydrolase; PON-1: paraoxonase-1; S1P: sphingosine 1-phosphate; SAA: serum amyloid A; T2DM: type 2 diabetes mellitus; TG: triglyceride; 🡫: decrease; 🡩: increase.

## Data Availability

Not applicable.

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
