# Peer review of "Dysfunctional High-Density Lipoproteins in Type 2 Diabetes Mellitus: Molecular Mechanisms and Therapeutic Implications"

_jcm, 2021, doi:10.3390/jcm10112233_

Round 1

Reviewer 1 Report

Authors have appropriately addressed the requested issues.

I have no more concerns.

Congratulations. Good job!

Reviewer 2 Report

The authors have responded satisfactorily to my requests. I recommend publication in current form.

Author Response

We acknowledge the reviewer for having appreciated our revision. 

This manuscript is a resubmission of an earlier submission. The following is a list of the peer review reports and author responses from that submission.

Round 1

Reviewer 1 Report

This review of HDL dysfunction in type 2 diabetes is an overall well written and comprehensive article. It adds to the growing literature in this area and clarifies the status as well as unknowns with regard to HDL in diabetes. While it doesn't break much new ground, it is a good concentrated source of recent information on this topic. However, there are some areas where it can be improved.

More major comments:

  • A critical emerging issue in HDL is the realization that the class represents a host of individual particles that perform distinct functions driven by different surface proteins. This is indirectly addressed in the narrative, but should probably be emphasized more with recent work that has shown the diversity of the HDL proteome and lipidome as well as evidence for specific subparticles relating to disease. For example, the recent work of Aroner and Sacks on the role of apoC-III containing HDL particles in T2D was not discussed. The addition of the HDL heterogeneity aspect would significantly strengthen the review and make it more useful for pointing the way forward.
  • It is not clear in Section 3.3 and 3.4 if the lipid and proteins changes cited in HDL are thought to be due to less particles (therefore less content) or remodeling of the particles themselves. The authors should  address this.
  • If the authors choose to include SGLT-2 inhibitors and its effects on HDL in the section of therapeutic interventions, there should be discussion of other anti-diabetes medications on HDL , ie.  metformin and GLP-1 agonists.
  • It would be worthwhile to add a section on paths moving forward or gaps in the current literature as determined by the authors

Minor Comments:

And Extra ‘and” is on page 1, line 31- it should be deleted

Introduction should clarify that this is a review of type 2 diabetes., not diabetes itself, page 2, line 46.

Section 3 second paragraph includes no references.  A reference should be added at line 100 and line 107

The term “diabetic subjects” is used in the text and the figure 1 title.  The correct term “patients with type 2 diabetes” should be used.

Reviewer 2 Report

In this review the authors examine the quantitative and qualitative abnormalities of high-density lipoproteins (HDL) in subjects with type 2 diabetes mellitus (T2DM) and their implications in the atheroprotective effects of these lipoproteins. Finally, they summarize the results of the main pharmacological intervention studies carried out for correcting diabetic dyslipidemia.

General comments

After a successful and obligatory introduction that addresses the increased cardiovascular risk in T2DM and the role of HDL in this clinical setting, the authors describe the phenotype of low HDL cholesterol and alterations in the composition and structure of HDL in diabetes. Apart from the contribution of HDL to effective reverse cholesterol transport, the authors discuss other non-classical functions such as antioxidant properties to modulate inflammatory responses, vasomotor reactions and blood clotting.

In the last section, they examine the main therapeutic interventions available to reduce cardiovascular risk in diabetes. Despite a robust association between low plasma HDL cholesterol levels and cardiovascular risk, therapeutic interventions aimed at increasing HDL cholesterol (e.g., niacin and CETP inhibitors) have turned out to be futile in outcome trials. I believe that this section does not add anything new; furthermore, it remains to be investigated whether therapies targeting quality of HDL could be effective in preventing progression of atherosclerosis and onset of cardiovascular disease in patients with T2DM.

There are many proofs regarding the connection between epigenetic factors and different diseases, including diabetes and cardiovascular disease. Interestingly, recent studies have shown that at least some anti-diabetic drugs exert epigenetic effects aside from their hypoglycemic. For this reason, I suggest that the authors could address this topic focusing on the available evidence with antidiabetic drugs.

Minor comments

I recommend including a table. For example, basic HDL proteins and their altered functions in T2DM.